# Enhanced polymer mechanical degradation through mechanochemically unveiled lactonization

Yangju Lin [1,2 ✉], Tatiana B. Kouznetsova[1], Chia-Chih Chang[1] & Stephen L. Craig [1,2 ✉]

The mechanical degradation of polymers is typically limited to a single chain scission per triggering chain stretching event, and the loss of stress transfer that results from the scission limits the extent of degradation that can be achieved. Here, we report that the mechanically triggered ring-opening of a [4.2.0]bicyclooctene (BCOE) mechanophore sets up a delayed, force-free cascade lactonization that results in chain scission. Delayed chain scission allows many eventual scission events to be initiated within a single polymer chain. Ultrasonication of a 120 kDa BCOE copolymer mechanically remodels the polymer backbone, and subsequent lactonization slowly (~days) degrades the molecular weight to 4.4 kDa, > 10× smaller than control polymers in which lactonization is blocked. The force-coupled kinetics of ring-opening are probed by single molecule force spectroscopy, and mechanical degradation in the bulk is demonstrated. Delayed scission offers a strategy to enhanced mechanical degradation and programmed obsolescence in structural polymeric materials.

[1] Department of Chemistry, Duke University, Durham, NC 27708, USA. [2]These authors jointly supervised this work: Yangju Lin, Stephen L. Craig.
✉email: yangju.lin@duke.edu; stephen.craig@duke.edu

Synthetic polymers have brought tremendous benefits to daily life, but the same structural stability that makes them so useful leads to a major challenge: they are often difficult to break down. Depending on the constituent polymers, disposed plastics persist for ~$10^2$–$10^3$ years in landfills and oceans with far reaching environmental consequences. Strategies for dealing with this challenge include biodegradable/degradable polymers[1,2], recyclable/reprocessable polymers[3–6], and polymers made from renewable sources[7]. The advance of degradable polymers has been facilitated by the tactical introduction of a rich range of biodegradable or stimuli-degradable motifs into the backbone of linear polymers and/or the cross-linkers of polymer networks[2,8–13], the degradation of which often occurs in the presence of an external stimulus (e.g., heat, light, chemical, electricity).

Nearly a century ago[14], mechanical forces were recognized as being capable of breaking down polymer chains and networks, including through the activation of intermolecular reactions such as oxidation[15]. But while mastication and other mechanical processing techniques are capable of triggering polymer degradation to an extent that changes mechanical properties, the utility of mechanical force for extensive polymer degradation is limited by the fact that the tension in a parent polymer chain relaxes after a single scission event (Fig. 1a), and the ability to recapture the chain in a high stress event drops with its molecular weight. In mechanical processing environments such as extrusion or sonication, therefore, polymer degradation plateaus at an apparent limiting molecular weight[16] that is not sufficiently low to meet desired degradation targets. Moore and Boydston et al.[17] reported a clever strategy to amplify a single triggering event through the scission-induced cascade depolymerization of a self-immolative cyclic poly(phthalaldehyde). This strategy, however, is not likely to be suitable for all degradation purposes, as the mechanically depolymerizable system relies on a metastable polymer backbone, and cascade degradation happens instantaneously upon activation, which might reduce stress transfer and limit the ability to

trigger events throughout all strands in a bulk material. We therefore sought a system in which mechanical force could trigger a delayed degradation, allowing polymer topology and mechanical properties to be retained so that many such initiating events could be induced. In particular, mechanophores[18] have been used in recent years to create a wide range of stress-coupled responses in polymer materials[18–21], including the extensive remodeling of polymers under processing conditions that might be suitable for inducing a large extent of degradation[19,22–26].

Inspired by these successes, we report here that mechanophores based on a [4.2.0]bicyclooctene (BCOE) core provide a delayed fragmentation response, which allows stress transfer to be maintained even as degradation events are being triggered. As a result, mechanical forces that would normally lead to a single scission event instead convert multiple copies of a mechanophore into products that undergo subsequent spontaneous but slow chain scission reactions. Specifically, mechanical activation of BCOE leads to a disrotatory, 4π-electron ring-opening reaction, the product of which is set up for spontaneous, intramolecular lactonization (Fig. 1b) that, over the course of days, leads to order-of-magnitude enhancements in the extent of mechanically triggered degradation relative to controls in which the lactonization is blocked.

## Results

**Thermodynamics of lactonization.** The BCOE mechanophore design is inspired by the beautiful work of Booker-Milburn et al.[27] and Ralph et al.[28] on the preparation of pachylactone from a hydroxy-substituted BCOE derivative **A** (Fig. 2a). When refluxed in xylenes, sequential ring opening and lactonization of **A** affords the target lactone product **E**. Based on prior work on mechanically triggered, disrotatory 4π-electron processes[29–32], we reasoned that a similar reaction cascade could be mechanically induced as long as the subsequent lactonization remains thermodynamically favorable. Computations supported the viability of ring-opened dihydroxy BCOE derivative **F**. As seen in Fig. 2b,

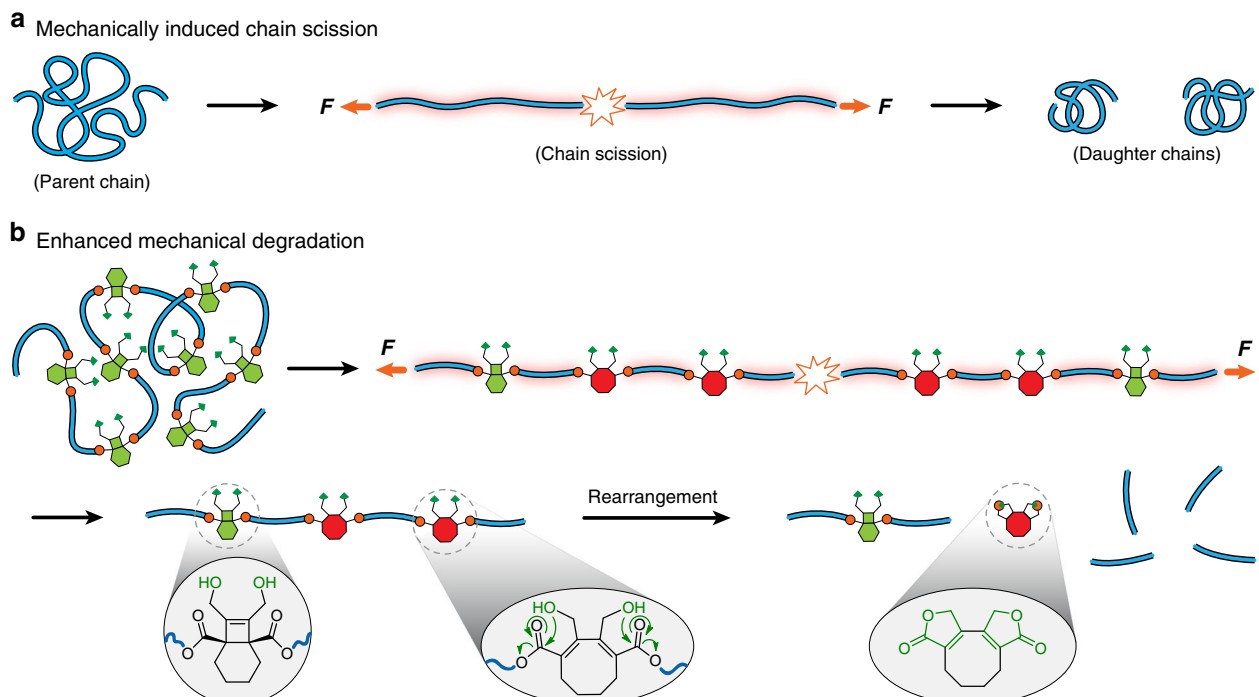

**Fig. 1 Illustration of mechanically induced chain scission and design of enhanced mechanical degradation. a** Subjection of polymer chain to mechanical force leads to chain scission at near midchain region and produces two daughter chains. **b** A polymer bearing BCOE mechanophores undergoes mechanically induced forbidden ring opening and subsequent lactonization rearrangement, leading to enhanced degradation.

the first intramolecular lactonization of **F** to **G** is exothermic by 9.4 kcal/mol, and a subsequent second lactonization to bis-lactone **H** releases an additional 2.8 kcal/mol. Based on these calculations, we hypothesized that polymers with the appropriate BCOE mechanophore precursor in the backbone would spontaneously release **H** following mechanical activation.

**Fig. 2 Synthesis of pachylactone from a BCOE derivative and thermodynamics of lactonization. a** Thermally induced sequential pseudo-forbidden ring opening and lactonization reported by Booker-Milburn and co-workers. **b** Computational study of the lactonization thermodynamics of a ring-opened BCOE species.

**Synthetic design**. The BCOE mechanophore was synthesized as shown in Fig. 3, using a slightly modified procedure based on reports by Booker-Milburn et al.[33–36]. BCOE anhydride **1** was prepared from the photochemical [2 + 2] cycloaddition of 3,4,5,6-tetrahydrophthalic anhydride and but-2-yne-1,4-diol. Subsequent acid-catalyzed esterification with but-3-en-1-ol gave the diene derivative **2**, of which the hydroxyls were protected with tetrahydropyran (THP) groups to yield **3**. Macrocycle **4** was obtained from ring-closing metathesis (RCM) of **3**, and **4** was further subjected to entropy-driven ring-opening metathesis polymerization (ED-ROMP) with co-monomer 9-oxabicyclo [6.1.0]non-4-ene (epoxy-COD) and 9,9-dichlorobicyclo[6.1.0] non-4-ene (gDCC-COD) to provide polymers **P1** ($M_n = 119$ kDa, $Đ_M = 1.48$, 26 mol% **4**) and **P2** ($M_n = 128$ kDa, $Đ_M = 1.39$, 14 mol% **4**), respectively. The epoxide-containing **P1** was employed in single molecule force spectroscopy (SMFS) studies, following previous reports in which the epoxide co-monomer was used to enhance adhesion to the cantilever[37]. Polymer **P2** was cleanly converted to **P3** ($M_n = 120$ kDa, $Đ_M = 1.62$, 14 mol% **4**) through the selective removal of THP-protecting groups (see Supporting Information). **P2** and **P3** were employed in sonication studies, with **P2** serving as a control for the role of lactonization, which is blocked by the presence of THP-protecting groups.

**Mechanical degradation**. The mechanical degradation of the polymers was investigated through pulsed ultrasonication of their tetrahydrofuran (THF) solutions. As shown in Fig. 4, ultra-sonication of **P2** leads to three mechanochemical outcomes: (1) a reduction in polymer molecular weight (MW), as indicated by the steady shift of gel permeation chromatography (GPC) traces to longer retention time (Fig. 4b); (2) the conversion of gDCC mechanophores into their respective 2,3-dichloroalkene products (as previously reported[38,39]); and (3) the ring opening of BCOE to the corresponding diene species. All three processes increase with increasing sonication time, but the rates of all gradually reduce with decreasing MW, as expected for mechanochemical processes. After 60 min of sonication, the MW of **P2** drops from 128 to 51 kDa, and this reduction is consistent with a typical outcome of polymer degradation under ultrasonication, where preferential force-induced scissions tend to occur in the chain-center region[40]. Over that same sonication time, the conversion of gDCC and BCOE mechanophores is 60% and 48%, respectively. This suggests that BCOE is similar to, but slightly less mechanically reactive

**Fig. 3 Synthetic design of BCOE mechanophore and preparation of polymers P1, P2, and P3.** pTSA refers to p-toluenesulfonic acid, and DCM is dichloromethane.

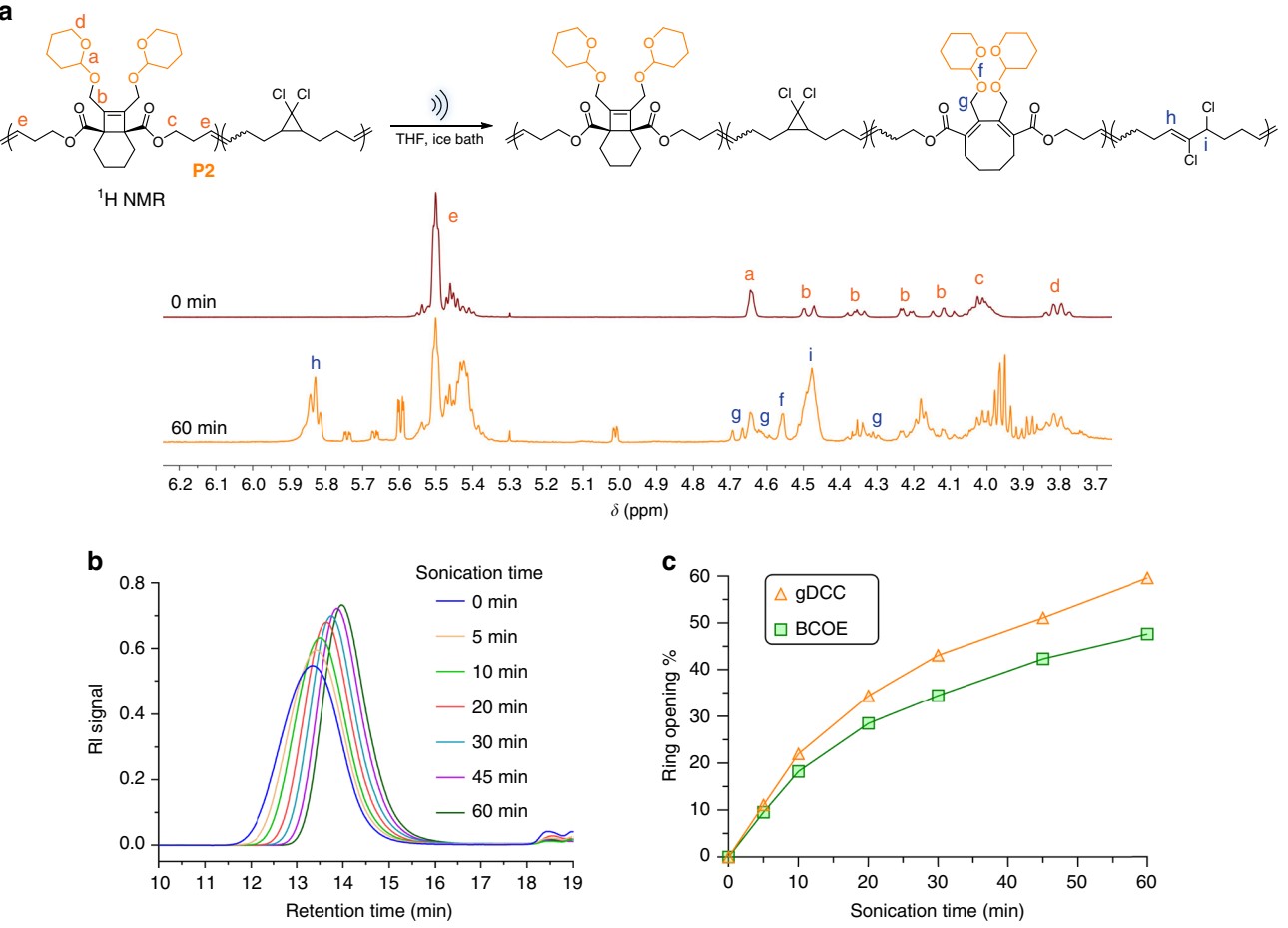

**Fig. 4 Ultrasonication study of polymer P2. a** Schematic presentation of ultrasonication induced mechanical activation of BCOE and $g$DCC in **P2**, and the $^1$H NMR characterization of activated species in **P2**. **b** Evolution of GPC traces of **P2** over ongoing sonication time. **c** Fraction of activated BCOE and $g$DCC at various sonication time. The fraction of activated BCOE was calculated using peak f, which corresponds to two protons in activated BCOE (Supplementary Fig. 3).

than, the well-known $g$DCC mechanophore. SMFS experiments confirm this conclusion (see below).

We anticipated similar extents of each of the three primary mechanochemical responses (homolytic chain scission, $g$DCC activation, and BCOE activation) in **P3** as were observed in **P2**, for the following reasons: (1) the rates of mechanochemical response are primarily determined by polymer contour length[41–43], which is effectively unchanged on going from **P2** to **P3**; (2) any differences in solvation of the BCOE alcohols and THP groups in THF have a negligible effect on sonochemical response, as suggested by a previous study;[43] and (3) the electrocyclic ring opening of the cyclobutene core is electronically decoupled from the OH/OTHP groups. As expected, proton nuclear magnetic resonance ($^1$H NMR) spectra confirm that the amount of activated $g$DCC internal standard at 60 min is effectively identical in **P3** (62%, Fig. 5a) to that observed for **P2** (60%), but slightly greater shifts in retention time in the associated GPC traces are observed for **P3** as a function of sonication time (Fig. 5d). These slight differences in shifted retention time suggest that small amounts of post-activation lactonization (and the associated chain scission) are occurring during the ongoing sonication and GPC analysis, and the $^1$H NMR spectra of sonicated **P3** reveal ~3% lactone after 60 min sonication and an additional 5 h standing time (Fig. 5b and Supplementary Fig. 12).

The post-sonication kinetics of lactonization were then studied in greater detail, using **P3** that had been sonicated for 60 min. The conversion of the lactone was quantified by integrating the $^1$H

NMR spectra, and it increases gradually over a couple of days until it plateaus at 49 ± 1%, within experimental uncertainty of the 48% BCOE ring opening observed in **P2** under identical sonication conditions. The production of the intended bis-lactone was further confirmed by mass spectroscopy (Supplementary Fig. 13). The generation of the lactone further correlates with an ongoing decrease in MW. As shown in Fig. 5c, the GPC traces of post-sonicated **P3** gradually evolve to reveal a bimodal distribution over ~45 h of standing time. This evolution in MW distribution continues for days and eventually reaches an apparent steady state after 221 h, in contrast to the 70 h observed in $^1$H NMR studies of lactonization. We attribute this three-fold difference in apparent lactonization rate to the different solvent environments of the two studies (i.e., $^1$H NMR experiments were conducted in CDCl$_3$, GPC experiments in THF).

The behavior of the system is consistent with the designed delayed-fracture mechanism, as each of two factors is required to be present simultaneously. First, the BCOE mechanophore must be opened to the corresponding diene, as pristine **P3** shows no evidence of spontaneous molecular weight degradation (Supplementary Fig. 11). Second, the diene product of ring opening must be able to undergo subsequent lactonization, as mechanically activated **P2**, which comprises of ring-opened BCOE but no unveiled hydroxy groups, shows no evidence of fragmented oligomers beyond single-event bond breaking in the GPC traces (Fig. 4b and Supplementary Fig. 7). Thus, we conclude that the intended scheme shown in Fig. 1 is indeed operative.

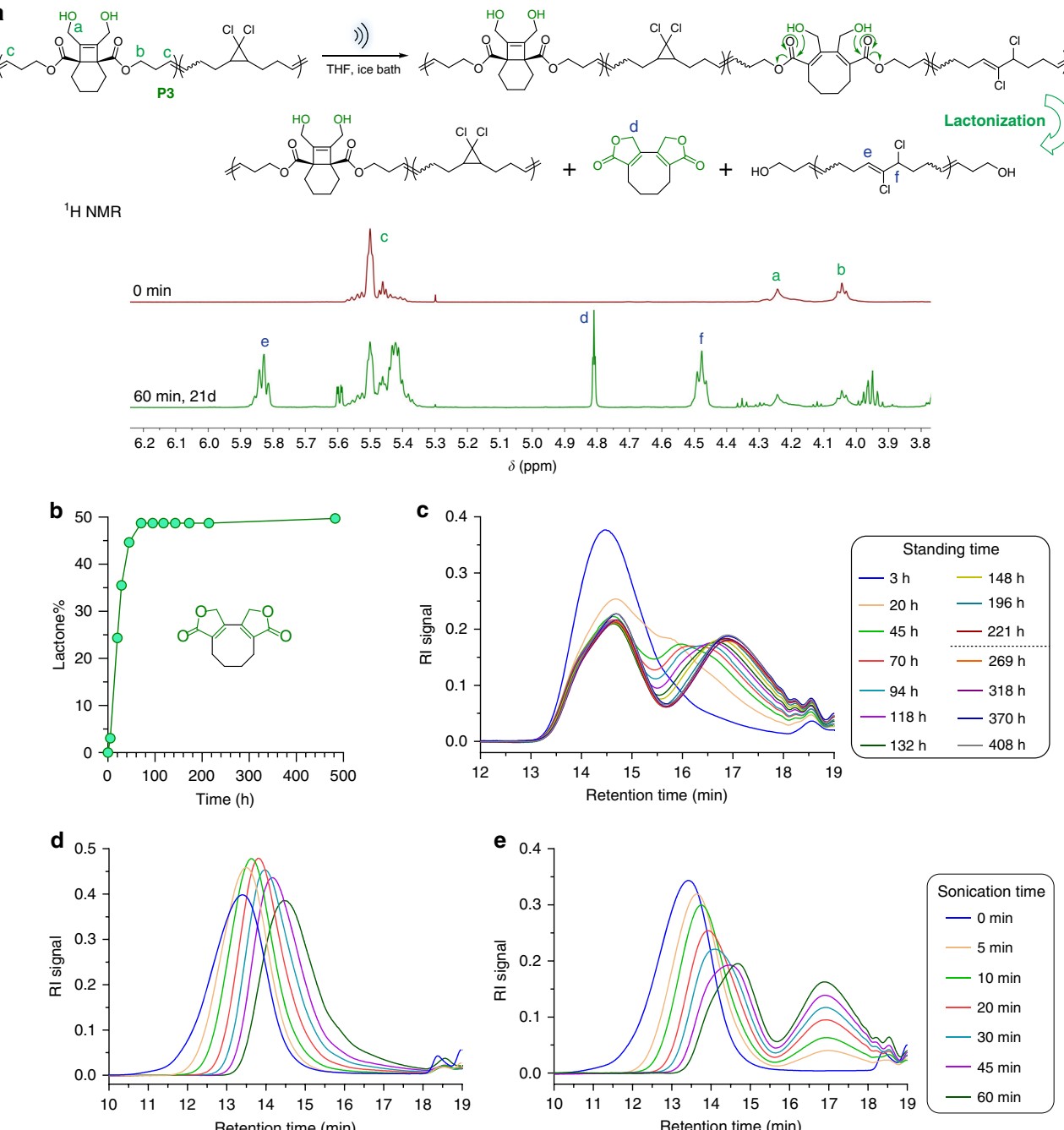

**Fig. 5 Ultrasonication study of polymer P3. a** Scheme of mechanical activation of BCOE and *g*DCC in **P3** and following spontaneous lactonization. [1]H NMR characterization of activated *g*DCC and generated lactone in a **P3** polymer that was sonicated for 60 min and set for 21 days to reach full lactonization. **b** [1]H NMR (CDCl₃) quantification of lactone formation over standing time in a **P3** polymer post 60 min sonication. **c** GPC trace evolution of **P3** polymer after 60 min sonication over various standing time in THF. GPC analysis of sonicated **P3** immediately after sonication (**d**) and after 17 days standing time (**e**). The legends for **d** and **e** are indicated on the right.

To ensure "complete" lactonization, each of the sonicated **P3** samples was allowed to sit under ambient conditions until a total of 17 days had passed since sonication. Subsequent characterization by GPC (Fig. 5e) shows a clean bimodal distribution, with a new and distinct low-MW peak centered at ~17 min retention time that increases as a function of initial sonication time. The observed bimodal distribution is consistent with the sonomechanochemical production of largely continuous blocks of activated and unactivated mechanophores, as reported in a previous system by Black Ramirez et al.[25] The MW of this peak, as determined by multi-angle static light scattering, is ~2 kDa. Coincident with the

growth in this oligomer peak, the higher MW peak decreases in intensity and gradually shifts to longer retention time. The extent of BCOE activation during sonication is limited by the uneven force distribution along the polymer backbone that is generated by the extensional flow fields produced in this experimental setup. Large forces that are sufficient for chain scission are produced near the center of the chain even as the ends of the chain are under modest tension, leaving BCOE in the terminal regions unactivated.

The MW of **P3** (combination of both peaks) following 60 min sonication and 17 days of subsequent lactonization has decreased

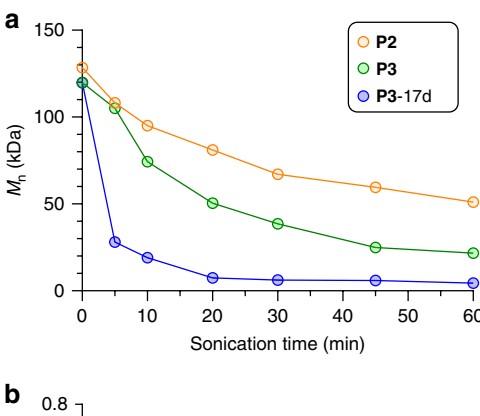

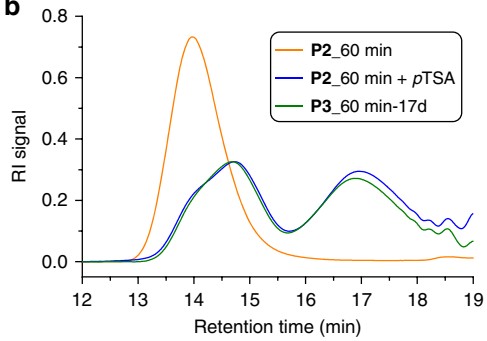

**Fig. 6 Mechanical degradation of P2 and P3, and mechanically regulated degradability of P2. a** Comparison of $M_n$ evolution over sonication time. Orange dots: **P2** polymer; blue dots: **P3** polymer analyzed immediately after sonication; green dots: **P3** polymer analyzed after sonication and further 17 days standing time. **b** GPC overlay of **P2** polymer after sonication, **P2** polymer after sonication and further $p$TSA treatment at room temperature overnight, and **P3** polymer after sonication and additional 17 days standing time. **P3**_60min-17d is normalized to the peak intensity of **P2**_60min $+p$TSA.

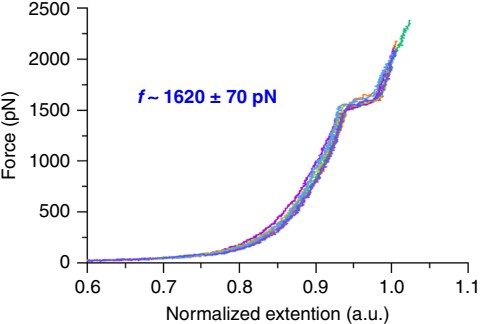

**Fig. 7 Representative force–extension curves of P1.** SMFS study was performed in toluene with a pulling velocity of 300 nm/s. Seven force–extension curves are normalized at the extension related to 2 nN force and overlaid in the figure.

from an initial value of 120 kDa to an ultimate value of 4.4 kDa. In contrast, the MW of the **P2** control plateaus at only 51 kDa (from an initial value of 128 kDa) under identical conditions (Fig. 6a), because the THP-protecting groups block subsequent lactonization. When $p$-toluenesulfonic acid is added to sonicated **P2**, it catalyzes the removal of the THPs and unveils the hydroxy groups. Following lactonization, the MW distribution of **P2** after sonication and further acid treatment is indistinguishable from that of sonicated **P3** (Fig. 6b). While used as a control experiment here, we note that the sequential ultrasonication and acid treatment of **P2** suggests a mechanically regulated degradability that complements previous designs of mechanically gated degradable polymers[24,26]. A key feature of the response in **P3** is that once the BCOE is activated, its subsequent lactonization proceeds slowly (time scale of days) and does not require additional mechanical input. Therefore, the lactonization rearrangement is the rate-limiting step in the chain fragmentation process. Because BCOE activation is non-scissile with respect to the polymer backbone, this allows many BCOE mechanophores to open prior to chain scission in ultrasonication (as many as ~32 BCOE per mechanically triggered chain scission event; see Supporting Information), and even greater levels of activation should be possible in chains under quasi-static load, as opposed to the extensional flow fields in ultrasonication.

**Single molecule force spectroscopy**. The single-chain response to quasi-static loading can be quantified using SMFS. SMFS was performed using techniques based on those reported previously by Craig and co-workers[30,37,44–46]. Representative force–extension

curves of **P1** are presented in Fig. 7. The characteristic plateau at average force of $\sim 1620 \pm 70$ pN is a consequence of rapid BCOE ring opening (force-coupled rate constant of $\sim 10^2\,s^{-1}$) along the polymer backbone, and the resulting extra extension ($6 \pm 2\%$) in the polymer chain is consistent with that calculated from CoGEF modeling (6%, see details provided in the Supporting Information). When the chains are stretched under constant tension via SMFS, in fact, the changes in polymer contour length are consistent with the quantitative activation of all BCOE mechanophores along the backbone prior to chain scission or detachment from the probe. Consistent with this observation, the force required for BCOE activation ($1570 \pm 40$ pN for a rate constant of $\sim 55\,s^{-1}$, Supplementary Fig. 28) is much lower than that required for homolytic scission of the backbone (likely 4–5 nN, based on extensive prior literature[39,47–49]). Thus, under conditions of quasi-static tension, such as those experienced in many bulk loading environments of elastomers, extensive remodeling of the backbone and high degrees of subsequent degradation are possible.

**Extrusion**. To demonstrate the utility of BCOE mechanophore in bulk degradation, we investigated the scission of polymer **P4** ($M_n =$ 69 kDa, $Đ_M = 1.71$, 5 mol% **4**) and **P5** ($M_n = 66$ kDa, $Đ_M = 1.72$, 5 mol% **4**) under extrusion, which is a common technique for plastic processing at an industrial scale. **P4** and **P5** were each blended with commercial polycaprolactone (PCL) at 2.5 wt% and subjected to extrusion (65 °C, 70 r.p.m., 2.1–0.8 MPa). Analysis of **P4** and **P5** within the blend is facilitated by the presence of chromophores in the polymer. Whereas the absorption of PCL at 332 nm is essentially undetectable by the in-line UV detector on the GPC, **P4** and **P5** absorb strongly at that wavelength due to the presence of a coumarin label (Supplementary Fig. 14). This labeling strategy allows for selective monitoring of the change in molecular weight of **P4** and **P5** within the majority PCL matrix. As seen in Fig. 8, the GPC trace of **P5** shifts to longer retention times that reflect significant molecular weight degradation, whereas the THP-protected control **P4** shows only a minimal change in the high molecular weight portion of the elugram (Supplementary Figs. 16 and 18). Thus, the degradation under extrusion is attributed to BCOE activation/lactonization, as opposed to random mechanical chain scission. In addition, control experiments at 65 °C in the absence of mechanical action results in no measurable activation in **P5** (Supplementary Fig. 21).

As expected, the kinetics of lactonization are influenced by the elevated temperature in the extruder. Unlike in the sonication experiments described above, the post-extrusion GPC traces remained effectively unchanged over the course of 10 days standing at room temperature (Figs. S15 and S17), indicating that

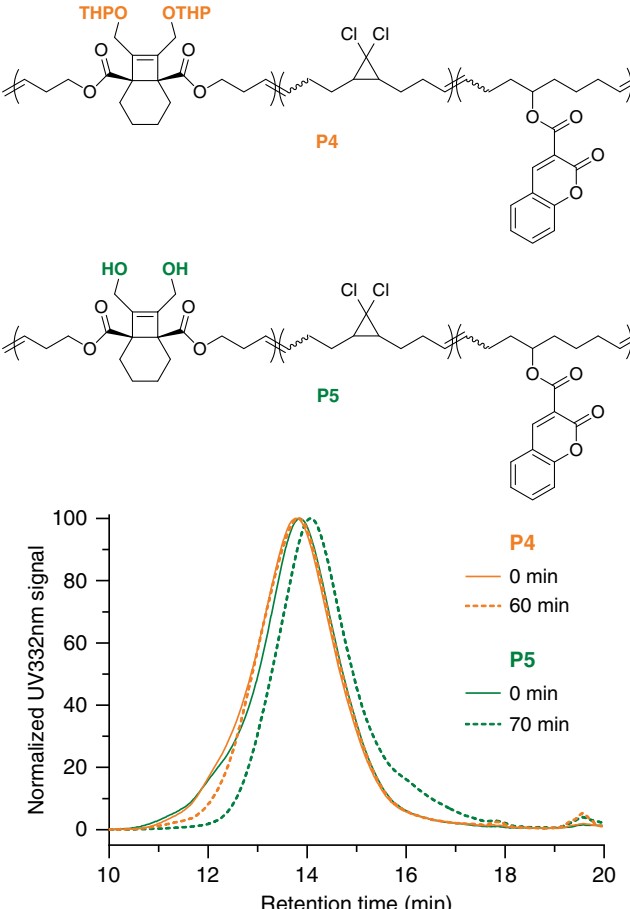

**Fig. 8 Extrusion induced degradation of P4 and P5. P4** and **P5** were blended into PCL matrix at 2.5 wt% ratio and subjected to extrusion. The UV absorbance at 332 nm (*y*-axis) is due to the coumarin co-monomer and allows for selective monitoring of **P4** and **P5** within the PCL matrix. After 10 days of standing time, **P4** (orange) and **P5** (green) polymers before (solid line) and after (dash line) extrusion were analyzed. The extrusion time for each polymer is indicated in the legend.

chain scissions to occur under conditions that would normally result in a single scission event. Preliminary bulk extrusion studies demonstrate that this and related strategies might find use in mechanically degraded polymers, such as in materials that upon extrusion or other mechanical working would degrade into much smaller components than would be possible if the fragmentation were limited to instantaneous single scission events. In addition, delayed degradation provides a potential route to materials with programmed obsolescence, in which a part subjected to early stage damage events would continue to function until its immediate tasks were completed, but then would subsequently fall apart in order to signal that a change was needed prior to redeployment. Finally, the kinetics of the unveiled degradation (here, lactonization) might be useful as time-dependent chemical signals whose evolution can be used to retroactively determine when a damaging event occurred, complementing a similar time stamping strategy reported recently[53]. Potential applications will require a careful balance of force-free stability, force-coupled activation kinetics, and desirable post-activation degradation kinetics, all of which must be tailored to the desired application. In the proof-of-concept system presented here, post-activation degradation can be tuned through the polarity and functionality of the polymer matrix, or, as shown here, increased temperature and/or the addition of a suitable catalyst. Whether through lactonization or entirely distinct designs, these opportunities motivate further studies into tandem reaction cascades that are triggered by a mechanical stimulus.

## Methods

**Sonication experiment**. Representative procedures. A solution of 34 mg polymer (**P2** or **P3**) in 17 mL dry THF was transferred into a dry Suslick cell. The solution was sparged with $N_2$ for 10 min under an ice bath. Pulsed ultrasound was applied (1 s on, 1 s off) at 30% amplitude. Aliquots of 0.8 mL sample with various sonication times were draw out for GPC analysis. Each of these samples was further condensed in a 10-mL scintillation vial and dried under high vacuum. Polymers were characterized by $^1$H NMR.

**SMFS**. Twenty microliters of a 0.05–0.1 mg/mL polymer solution was added to the silicon substrate surface and allowed to dry. The silicon substrate was then placed on the piezoelectric stage of the atomic force microscopy (AFM), and toluene was added to solvate the deposit polymers. Then, the AFM tip was brought into contact with the substrate and then retracted at a velocity of 300 nm/s. The approaching/retracting cycles were recorded in dSPACE (dSPACE Inc. Wixom, MI) and analyzed using Matlab (The MathWorks, Inc., Natick, MA).

**Extrusion**. Extrusion studies were performed on a HAAKE™ MiniCTW Micro-Conical Twin Screw Compounder. The compounder comprises a clamshell barrel with two conical screws and a recirculation pathway. The barrel was preheated to 65 °C and the screw rotation were set to 70 r.p.m. Polymer pieces (3.1 g) were then added in portions using a mechanical plunger. Samples were removed for analysis after various extrusion times.

## Data availability

Raw data sets, including NMR spectra, GPC-MALS data, SMFS data, and extrusion data, that were generated and/or analyzed as part of this study are available from https://doi.org/10.7924/r4fq9x365.

the lactonization is nearly completed during extrusion. At the same time, the extent of degradation increases with extrusion time (Supplementary Fig. 20). In contrast to the slow lactonization kinetics post-sonication, the observed complete lactonization after extrusion is ascribed to the elevated temperature (65 °C) during extrusion that facilitates the cascade lactonization. While extent of degradation observed here is significantly less than observed in the sonication experiments, in principle it can be increased through processing conditions and times that are optimized in the context of a given material. In that regard, the observation of fairly low molecular degradation fragments (<10 kDa, Supplementary Fig. 22 and Section I.3.5) is promising. We note that full degradation is not always necessary, as in some cases partial degradation can increase the biodegradability of high polymers[50–52]. Such optimization should also balance the desire for activated mechanophores that remain intact for long enough to maintain effective stress transfer for additional mechanophore activation, while also degrading quickly enough to be useful.

## Discussion

The strategy reported here, of delayed degradation enabled by mechanochemically unveiled functionality, stands in contrast to existing mechanical degradation schemes in that it enables multiple

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

## Acknowledgements

This material is based on work supported by the National Science Foundation under grant CHE-1808518 (to S.L.C.). The authors acknowledge the Duke Mass Spectroscopy Center and Dr. Peter Silinski for mass spectroscopy and subsequent analysis.

## Author contributions

Y.L. and C.-C.C. conceived the idea and designed the experiments. Y.L. performed the calculation, synthesis, and sonication experiments. T.B.K. performed the SMFS experiments and collected the data. Y.L., C.-C.C., and S.L.C. analyzed the data and wrote the manuscript.

## Competing interests

The authors declare no competing interests.
