## [Peer Review File · Nature Communications]

REVIEWER COMMENTS

Reviewer #1 (Remarks to the Author):

The authors report here the elegant design of a BCOE mechanophore allowing the non-scissile mechanochemical ring-opening of multiple BCOE moieties along a single polymer chain with a subsequent delayed and mechanochemically gated lactonization leading to the degradation of the respective polymer. This strategy is interesting and loosely related to a recent work by Craig and coworkers (Ref. 19) and Wang and coworkers (Ref. 17). The manuscript is well and instructively written, the experimental data are of high quality, and the conclusions drawn from them appear justified. Considering the urgency to solve polymer waste production by creative chemical approaches, this manuscript surely has potential high impact and a wide scope appealing to the broad readership of Nature Communications. Hence, I would be inclined to recommend publication however considering the, in my opinion, following shortcomings.

The authors end the main body of text with the sentence: "Thus, under conditions of quasi-static tension, such as those experienced in many bulk loading environments of elastomers, extensive remodeling of the backbone and high degrees of subsequent degradation are possible." But this is where this research becomes exciting, relevant, and impactful. Here, the decisive experiments, at least in a limited fashion, should be presented. The question is, how, and how well the BCOE mechanophore responds to force application (extrusion, mastication) in commercial-like materials, such as (filled) rubbers or simple bulk batches/blends must be addressed. An, at least, model-wise large-scale useful form of mechanical degradation must be shown to be compatible with BCOE-equipped polymer materials in bulk. No commercial process will ever rely on dilute THF solutions in conjunction with sonicators to degrade polymers! Sonication of polymer solutions can only be used as a laboratory model for generating "force" and produces considerably higher shear rates compared to bulk methods. If the available amount of BCOE-containing polymer in the authors' labs is too little, I suggest (solution-) blending it with a commercial miscible polymer, such as poly(norbornadiene), and using NMR after bulk force application and dissolution to prove bulk mechanophore activation and subsequent degradation. Mastication should work, extrusion will most likely be hard due to the temperature sensitivity of the BCOE moiety.

Moreover, on p. 2, ll. 54 the authors state that while the mechanochemically-induced ring-opening reaction proceeds more or less instantaneous on force application, the lactonization-induced degradation occurs over multiple days. I understand that mechanochemical activation and degradation need to be temporally separated for the sake of quantitative mechanophore activation, but multiple days seem rather long from a technical and logistic perspective when scaled up. Could the authors briefly discuss the principal technical possibilities and limitations? This is chemistry and not process engineering, so I am not expecting a prototype layout but a perspective on feasibility is in order, I believe.

These issues crucially need to be addressed before publication in Nature Communications because they form the major motivation for developing mechanically degradable polymers. Unfortunately, this cannot be done entirely without new experiments, but I hope the authors find a way to still realize this in these challenging times. Without these experiments, this manuscript unfortunately adds too little progress to the already existing work in this field (Ref. 17 and 19).

In addition, please note the following points in the order of their appearance:

Manuscript:

1. L. 75: "4-buten-1-ol" should most likely be "but-3-en-1-ol" following basic nomenclature rules.
2. L. 80: Dispersity of molar mass (I assume) should be indicated as such with \overline{M}_w instead of only \overline{M} .
3. Ll. 80: The authors synthesize the copolymer P1 including epoxide comonomer for cantilever adhesion during SMFS.
 - 3.1. The characterization data, i.e. molar mass and dispersity in molar mass, should be given analogously to P2/P3.
 - 3.2. the epoxide moiety was shown by Craig and coworkers to be a mechanophore, though a comparably bad one. See: *Macromolecules* 52, 6234–6240 (2019); *J. Am. Chem. Soc.* 134, 9577–9580 (2012). Does this affect characterization via SMFS and comparability of the results?

4. L. 90: GPC as abbreviation should be written fully once.
5. L. 94: "drop off" is jargon.
6. L. 100: abbreviation for "single molecule force spectroscopy" was already introduced above.
7. L. 109: 1H NMR as abbreviation should be written fully once.
8. L. 168: SMFS abbreviation was introduced above.
9. L. 203: AFM as abbreviation should be written fully once.
10. L. 210: "rotavaped" is jargon.
11. L. 211: "ml" should be "mL".
12. L. 211: "polymers" should be capitalized.
13. LI. 253: Reference 17 is missing issue and page numbers.
14. LI. 296: Reference 33 has a spelling error.

Supplementary Information:

1. L. 40: "3,4-Dihydro-2H-pyran" should not be capitalized.
2. L. 40: the p in "pTSA" should be italicized.
3. L. 48: Either "hertz" or "Hz".
4. L. 49: If the authors really favor double blind peer review, they should remove fragments, such as "Duke University's Mass Spectrometry Facility" in the future.
5. L. 157: "PDI" is used here and sometimes below, "D" in the manuscript and other SI sections.
6. L. 162: Yield missing.
7. L. 170 "mL" not "ml".
8. General: it would be good if the authors could add a y-axis to their GPC elugrams and indicate whether spectra were normalized.

Reviewer #2 (Remarks to the Author):

The authors reported a very interesting mechanophore design based on hydroxy-substituted [4.2.0]bicyclooctene (BCOE), which undergoes facile mechanochemical 4n electrocyclic ring-opening to yield a cyclooctadiene based intermediate. The intermediate exhibits a favored geometry to undergo subsequent lactonization to cleave the ester linkages in polymer backbone and results in polymer degradation. The force-coupled polymer degradation process was systematically characterized by NMR, GPC, SMFS and DFT calculations. The clever design enables polymers with controlled and enhanced mechanical degradation profiles beyond the simple classic polymer mechanical cleavage that occurs in the middle of the chain. I recommend publication of this manuscript after minor revisions:

1. some GPC traces in the manuscript don't seem normalized to the same concentration (such as Fig 4b, Fig 5d). were the GPC samples taken at the same concentration? Careful normalization of intensity will help estimate the relative population of different polymer species.
2. Figure 5c GPC shows P3 after activation becomes a bimodal distribution overtime as the lactonization slowly cleaves the polymer backbone. However, the higher MW peak around 14-15 min seems to have negligible changes in the elution time. If the lactonization continuously occurs along the backbone, I would expect this peak to shift to longer elution times. Any explanation?
3. NMR of THP protected alcohols is complex and may interfere with the analysis of activation. some reference will help to ensure all the peak assignment. Using TMS protecting group (in the future) could keep the NMR simple.
4. In figure 6a, the blue and green circles wer mislabeled.
5. For the GPC trace of P2_60min+pTFA in figure 6b, how long did the sample stand before GPC measurement? Does TFA also catalyze the lactonization process?
6. In the SMFS section, can the authors comment how the force of activation compares to other cyclobutene mechanophores? Does the bicyclic structure lead to higher or lower force for ring-opening?

Reviewer #3 (Remarks to the Author):

This manuscript describes an interesting way to enable mechanochemically-triggered, delayed polymer degradation through a clever intramolecular lactonization strategy. The authors utilize a novel BCOE monomer with alcohol substituents (following deprotection) capable of lactonization following mechanochemical ring opening.

While the concept is interesting, the manuscript very clearly and concisely written, and the experimental work is performed at a very high level, there are some drawbacks of this paper that mostly pertain to bigger picture questions and novelty (see comments below). In any case, this work is interesting and could be published in Nature Communications following revision.

Specific comments:

- The concept of damaging a polymer to enhance its degradation is not particularly novel. The authors have shown a nice way of doing that with a very clear mechanism and a well-defined, novel model system. Nevertheless, it would be helpful for the general audience to discuss how this concept has been achieved using other strategies. Even in classical polymers/plastics, mechanical degradation can enhance rates of oxidation and eventual degradation (through, for example, increasing surface area). Additionally, shorter polymers are sometimes more susceptible to biodegradation; thus, an initial cleavage event can enhance the rate of subsequent biodegradation.
- Some discussion of practicality of this approach would be relevant. There are many very clever examples of installing mechanophores into polymers to induce interesting force-driven transformations, and while most of these are great from a fundamental science/academic perspective, their potential to impact real-world applications remains unclear. In this paper, where the authors cast their work in the context of sustainability, one must ask if the cost/synthesis/degradation products/etc. could make these materials (or this strategy) sustainable?

I realize that question cannot be fully answered now, but any hints/suggestions of how the authors see this work translating into the future would be great to add if possible. Perhaps strategies like this could be used to generate additives that somehow trigger degradation of a matrix of lower-cost material?

- Further in terms of novelty, this paper has some similarities to references 17 and 19, where mechanochemical cleavage leads to polyesters that can degrade over longer timescales. While the mechanism used here is clearly different, given the complexity introduced I am not sure it is advantageous.
- Page 1, line 27: "facilitated by the tactical introduction of a rich range of biodegradable or stimuli-degradable motifs into the polymer backbone." This phrase should have associated references. Additionally, in most cases, the motifs are either esters (as used in this work) or amides or they are often not in the "polymer backbone" but instead between polymer strands (for example, in many CANs). Perhaps this distinction could be clarified?
- Page 2, line 50: The references selected for 15-19 seem a bit arbitrary. Especially given that the sentence mentions "large scale degradation"? Some of the cited examples use monomers that are unlikely to be amenable to large scale processes (unless here the authors mean something different by large scale degradation (such as most of the bonds of a single macromolecule)? If so, the wording is unclear).
- Page 2, line 62 (and Figure 2): Isn't the process shown in Figure 2a conrotatory? I.e., the example from Booker-Milburn?

- Page 2, line 67: "gains an additional 2.8 kcal/mol." Should this be changed to "releases an additional 2.8 kcal/mol"? As written, it suggests an endothermic process to me.
- Page 2: synthetic design: mentioned reaction yields here. They are provided in the SI, but would be helpful in the main text (or in Figure 3).
- Page 4, line 171: "P1 is" should be "P1 are"
- Perhaps I overlooked this somewhere in the manuscript: was it necessary to ROMP the protected monomer? I.e., would ROMP of the deprotected monomer not work?
- Figure S4: I am not sure what "left" and "right" are referring to.
- How confident are the ¹H NMR assignments shown, for example, in Figure 4a? For example, the large shift of proton a to f?
- This paper would be much more compelling if degradation of bulk samples was demonstrated. The authors should mention why such studies were not or could not be conducted.

REVIEWER COMMENTS

Reviewer #1:

Comment. The authors end the main body of text with the sentence: “Thus, under conditions of quasi-static tension, such as those experienced in many bulk loading environments of elastomers, extensive remodeling of the backbone and high degrees of subsequent degradation are possible.” But this is where this research becomes exciting, relevant, and impactful. Here, the decisive experiments, at least in a limited fashion, should be presented. The question if, how, and how well the BCOE mechanophore responds to force application (extrusion, mastication) in commercial-like materials, such as (filled) rubbers or simple bulk batches/blends must be addressed. An, at least, model-wise large-scale useful form of mechanical degradation must be shown to be compatible with BCOE-equipped polymer materials in bulk. No commercial process will ever rely on dilute THF solutions in conjunction with sonicators to degrade polymers! Sonication of polymer solutions can only be used as a laboratory model for generating “force” and produces considerably higher shear rates compared to bulk methods. If the available amount of BCOE-containing polymer in the authors’ labs is too little, I suggest (solution-) blending it with a commercial miscible polymer, such as poly(norbornadiene), and using NMR after bulk force application and dissolution to prove bulk mechanophore activation and subsequent degradation. Mastication should work, extrusion will most likely be hard due to the temperature sensitivity of the BCOE moiety.

Response. We acknowledge that this is a weakness of the original submission. Following the reviewer’s suggestion, we have blended a UV-tagged, BCOE polymer with polycaprolactone (PCL) and subjected it to extrusion. The required temperature of 65 °C was found to be low enough for the critical experiments. Namely, we find that we can degrade the BCOE polymer, and that the cascade scission of the mechanophore results in lower molecular weights (including parts of the MW distribution as low as ~few kD) than are obtained from any instantaneous chain scission (which appears to be minimal). We have added a new section to the results and discussion, and we have amended the conclusion as well to reflect these data.

Comment. Moreover, on p. 2, ll. 54 the authors state that while the mechanochemically-induced ring-opening reaction proceeds more or less instantaneous on force application, the lactonization-induced degradation occurs over multiple days. I understand that mechanochemical activation and degradation need to be temporally separated for the sake of quantitative mechanophore activation, but multiple days seem rather long from a technical and logistic perspective when scaled up. Could the authors briefly discuss the principal technical possibilities and limitations? This is chemistry and not process engineering, so I am not expecting a prototype layout but a perspective on feasibility is in order, I believe.

Response: Fair point, and certainly one we have considered. The delayed lactonization kinetics of this precise mechanophore are tunable through temperature, solvent, and the addition of catalysts. In addition, changes in mechanophore structure (most notably, ring strain) can also be used. Related comments have been added in the conclusion section. It is worth noting that even this mechanophore appears to react quite quickly under the conditions of extrusion we employ in the bulk material studies.

Comments. In addition, please note the following points in the order of their appearance:
Manuscript:

1. L. 75: “4-buten-1-ol” should most likely be “but-3-en-1-ol” following basic nomenclature rules.
2. L. 80: Dispersity of molar mass (I assume) should be indicated as such with \mathcal{D}_M instead of only \mathcal{D} .
3. Ll. 80: The authors synthesize the copolymer P1 including epoxide comonomer for cantilever adhesion during SMFS.

3.1. The characterization data, i.e. molar mass and dispersity in molar mass, should be given analogously to P2/P3.

Response: All adjusted as suggested.

Comment. 3.2. the epoxide moiety was shown by Craig and coworkers to be a mechanophore, though a comparably bad one. See: *Macromolecules* 52, 6234–6240 (2019); *J. Am. Chem. Soc.* 134, 9577–9580 (2012). Does this affect characterization via SMFS and comparability of the results?

Response: No. The epoxide could not be activated at force up to 2.5 nN under SMFS condition, as has been indicated in previous study (*Nat. Chem.* 2013, 5 (2), 110-114). The mechanical activation of epoxide was realized through backbone level-arm (*J. Am. Chem. Soc.* 2012, 134 (23), 9577-80.) or in an allylic epoxide derivative (*Macromolecules* 2019, 52 (16), 6234-6240.) under ultrasonication, but even in these more active cases we have not observed any epoxide activity up to 2.5 nN of force, well above the forces at play here.

Comments.

4. L. 90: GPC as abbreviation should be written fully once.

5. L. 94: “drop off” is jargon.

6. L. 100: abbreviation for “single molecule force spectroscopy” was already introduced above.

7. L. 109: ¹H NMR as abbreviation should be written fully once.

8. L. 168: SMFS abbreviation was introduced above.

9. L. 203: AFM as abbreviation should be written fully once.

10. L. 210: “rotavaped” is jargon.

11. L. 211: “ml” should be “mL”.

12. L. 211: “polymers” should be capitalized.

13. Ll. 253: Reference 17 is missing issue and page numbers.

14. Ll. 296: Reference 33 has a spelling error.

Response: All revised accordingly

Comments.

Supplementary Information:

1. L. 40: “3,4-Dihydro-2H-pyran” should not be capitalized.

2. L. 40: the p in “pTSA” should be italicized.

3. L. 48: Either “hertz” or “Hz”.

Response: All revised accordingly

Comment. 4. L. 49: If the authors really favor double blind peer review, they should remove fragments, such as “[redacted]’s Mass Spectrometry Facility” in the future.

Response: Ugh. Thanks for pointing this out. We obviously missed this one (we wrote the manuscript first and decided on double-blind later), and it has been removed for further review (although it is likely that any referees that missed it during the review will now have seen it in the comments).

Comments.

5. L. 157: "PDI" is used here and sometimes below, "Đ" in the manuscript and other SI sections.

6. L. 162: Yield missing.

7. L. 170 "mL" not "ml".

Response: Revised accordingly.

Comment. 8. General: it would be good if the authors could add a y-axis to their GPC elograms and indicate whether spectra were normalized.

Response: The y-axis has been added and related description has been detailed in the caption.

Reviewer #2:

Comment. 1. some GPC traces in the manuscript don't seem normalized to the same concentration (such as Fig 4b, Fig 5d). were the GPC samples taken at the same concentration? Careful normalization of intensity will help estimate the relative population of different polymer species.

Response: For a specific polymer with various sonication times, each GPC sample was taken and analyzed at the same concentration. The reason the traces don't appear to be normalized is that the peak shape (width) is changing – the area is effectively constant.

Comment. 2. Figure 5c GPC shows P3 after activation becomes a bimodal distribution overtime as the lactonization slowly cleaves the polymer backbone. However, the higher MW peak around 14-15 min seems to have negligible changes in the elution time. If the lactonization continuously occurs along the backbone, I would expect this peak to shift to longer elution times. Any explanation?

Response: Yes, actually this makes sense. What is happening (or at least should be happening) is that initially after sonication, there is a long stretch of unactivated polymer connected to stretches of activated polymer (see ACS Macro Lett., **2012**, *1*, 23-27). Now imagine that half of the activated mechanophores lactonize. This leaves a high MW "unactivated" polymer with 0, 1, or 2 activated mechanophores dangling off the end, and a bunch of lower MW fractions with activated mechanophores in the middle. Subsequent lactonization does little to change the MW of the high MW fractions, but would still cut the low MW fractions in half. What we observe here is consistent with this expectation.

Comment. 3. NMR of THP protected alcohols is complex and may interfere with the analysis of activation. some reference will help to ensure all the peak assignment. Using TMS protecting group (in the future) could keep the NMR simple.

Response: Thank you for your suggestion. We have also performed a COSY to aid the initial peak assignments.

Comment. 4. In figure 6a, the blue and green circles were mislabeled.

Response: We have checked the figure again, they are correct.

Comment. 5. For the GPC trace of P2_60min+pTFA in figure 6b, how long did the sample stand before GPC measurement? Does TFA also catalyze the lactonization process?

Response: The sample was set at room temperature for overnight. Yes, pTSA does catalyze and accelerate the lactonization. Description of P2_60min+pTSA has been detailed in the caption.

Comment. 6. In the SMFS section, can the authors comment how the force of activation compares to other cyclobutene mechanophores? Does the bicyclic structure lead to higher or lower force for ring-opening?

Response: The SMFS quantification of cyclobutene has not been reported yet, although we do have some unpublished SMFS results of some cyclobutene derivatives. The fused bicyclic structure does not change the SMFS curve (at these pulling rates) significantly relative to unfused cyclobutene derivatives (within 100-200 pN). A discussion of these relationships in this paper seems out of scope, however.

Reviewer #3:

Specific comments:

- The concept of damaging a polymer to enhance its degradation is not particularly novel. The authors have shown a nice way of doing that with a very clear mechanism and a well-defined, novel model system. Nevertheless, it would be helpful for the general audience to discuss how this concept has been achieved using other strategies. Even in classical polymers/plastics, mechanical degradation can enhance rates of oxidation and eventual degradation (through, for example, increasing surface area). Additionally, shorter polymers are sometimes more susceptible to biodegradation; thus, an initial cleavage event can enhance the rate of subsequent biodegradation.

Response: Nice point. We have added mentions of these strategies in relevant places in the manuscript.

- Some discussion of practicality of this approach would be relevant. There are many very clever examples of installing mechanophores into polymers to induce interesting force-driven transformations, and while most of these are great from a fundamental science/academic perspective, their potential to impact real-world applications remains unclear. In this paper, where the authors cast their work in the context of sustainability, one must ask if the cost/synthesis/degradation products/etc. could make these materials (or this strategy) sustainable?

Response: As discussed in response to comments by Reviewer #1, we acknowledge that the absence of a connection to bulk materials processing is a weakness of the original submission. In response, we have blended a UV-tagged, BCOE polymer with polycaprolactone (PCL) and subjected the blend to extrusion. We find, and have added to the revised manuscript, that we can degrade the BCOE polymer, and that the delayed scission of the mechanophore results in lower molecular weights (including parts of the MW distribution as low as ~few kD) than are obtained from any instantaneous chain scission (which appears to be minimal).

- I realize that question cannot be fully answered now, but any hints/suggestions of how the authors see this work translating into the future would be great to add if possible. Perhaps strategies like this could be used to generate additives that somehow trigger degradation of a matrix of lower-cost material?

Response: Some acid/catalyst generating polymers in response to mechanical load could serve well for this purpose and can be found elsewhere (J. Am. Chem. Soc. 2012, 134 (30), 12446-12449.; J. Am. Chem. Soc. 2016, 138 (8), 2540-3.; J. Am. Chem. Soc. 2020, 142 (1), 99-103; *Poly. Chem.* **2013**, 4 (18), 4846-4859.).

- Further in terms of novelty, this paper has some similarities to references 17 and 19, where mechanochemical cleavage leads to polyesters that can degrade over longer timescales. While the mechanism used here is clearly different, given the complexity introduced I am not sure it is advantageous.

Response: In contrast to the ref 17 and 19 that use mechanical force to regulate the polymer degradability, in which the polymer degrades in the presence of both mechanical stimulus and degrading trigger. The enhanced mechanical degradation reported here takes advantage of the cascade, delay lactonization, which allow polymer to fragment under mechanical force.

- Page 1, line 27: “facilitated by the tactical introduction of a rich range of biodegradable or stimuli-degradable motifs into the polymer backbone.” This phrase should have associated references. Additionally, in most cases, the motifs are either esters (as used in this work) or amides or they are often not in the “polymer backbone” but instead between polymer strands (for example, in many CANs). Perhaps this distinction could be clarified?

Response: References have been added. Both linear polymer and polymer network have been addressed.

- Page 2, line 50: The references selected for 15-19 seem a bit arbitrary. Especially given that the sentence mentions “large scale degradation”? Some of the cited examples use monomers that are unlikely to be amenable to large scale processes (unless here the authors mean something different by large scale degradation (such as most of the bonds of a single macromolecule)? If so, the wording is unclear).

Response: We really meant “extensive degradation” in terms of breaking a higher fraction of polymer chains. The description has been clarified.

- Page 2, line 62 (and Figure 2): Isn’t the process shown in Figure 2a conrotatory? I.e., the example from Booker-Milburn?

Response: Yes, it’s conrotatory. Corrected in the main text.

- Page 2, line 67: “gains an additional 2.8 kcal/mol.” Should this be changed to “releases an additional 2.8 kcal/mol”? As written, it suggests an endothermic process to me.

Response: Corrected.

- Page 2: synthetic design: mentioned reaction yields here. They are provided in the SI, but would be helpful in the main text (or in Figure 3).

Response: The reaction yields have been added in Figure 3.

- Page 4, line 171: “P1 is” should be “P1 are”

Response: Corrected.

- Perhaps I overlooked this somewhere in the manuscript: was it necessary to ROMP the protected monomer? I.e., would ROMP of the deprotected monomer not work?

Response: Theoretically, ROMP of deprotected monomer would work as well. We had a concern that allylic alcohol may affect the ROMP, and so we decided to try this post polymerization deprotection. In addition, this post polymerization modification allows for direct comparison of the same polymer chain.

- Figure S4: I am not sure what “left” and “right” are referring to.

Response: It is a typo and has been corrected now.

- How confident are the ¹H NMR assignments shown, for example, in Figure 4a? For example, the large shift of proton a to f?

Response: We’ve performed COSY on the starting material, see page S30 in revised SI. The assignment of peak f in the ring opened product is, we think, likely, but we admit potential ambiguity. Nonetheless, the key use of f is not its assignment, but that its integration corresponds to 2 protons as used in the % activation calculations. We have verified those calculations by removing the THP groups with acid and repeating the ¹H NMR analysis, as described on page S9.

- This paper would be much more compelling if degradation of bulk samples was demonstrated. The authors should mention why such studies were not or could not be conducted.

Response: As mentioned above, we have added the extrusion study, and the results of bulk degradation has been discussed.

REVIEWERS' COMMENTS:

Reviewer #1 (Remarks to the Author):

In the revision of their original manuscript, the authors performed a serious amount of additional experiments regarding the initially criticized lack of bulk mechanochemical degradation proof of concept. They used a commercial extruder and blended their mechanochemically responsive BCOE-containing polymers with polycaprolactone, which is fair due to the large amounts that extruders use up. The authors added control experiments showing that the THP-protected polymer does not respond with degradation to the extrusion process and thermal controls show that while the lactonization kinetics, and thus chain scission, are accelerated by heat in the extruder, the initial mechanochemical ring-opening process remained unaffected. Cleverly, the authors use a dye-labelled polymer so the BCOE polymers can be selectively followed by GPC without PCL interference. With these results, the authors clearly demonstrate the feasibility of the degradation process in the bulk and hence give a very promising outlook on the implementation of such a technology in commodity polymers. Though degradation in bulk did not reach the extensive character as compared to the sonication experiments (yet), I agree with the authors here that this is merely a technical issue and I believe such time-consuming iterative optimizations have no place within fundamental cutting-edge research. Overall, I am very much convinced by the excellent work of the authors that addressed all of the issues raised by me earlier and strongly suggest accepting this manuscript for publication in Nature Communications.

Robert Göstl

Reviewer #2 (Remarks to the Author):

the authors have properly addressed the comments and this manuscript can be published.

Reviewer #3 (Remarks to the Author):

I appreciate the authors' efforts in revising their manuscript. I am not sure how meaningful the new extrusion experiment is: (1) if the polymer degrades upon typical processing, would it be useful? (2) in what application would adding 2.5 wt% of something that degrades be useful? Nevertheless, it is better than nothing.

I also found some of the responses to be less than ideal. For example:

- 1) In response to my comment about the practicality of the approach and casting the work in the context of sustainability, the authors reference a response to a different reviewer focused on bulk material degradation. That is not really what I was asking about, and if anything the extrusion experiment conducted drives home questions about the relevance of this approach.
- 2) The response to my next comment about how this work could translate in the future cites other papers but does not answer the question directly. If anything, citing other works detracts from the current work, showing that previous papers have addressed how this might be useful.
- 3) In response to my comment about novelty and the advantages of this mechanism, the authors restate the obvious. Notably, however, their statement that the degradation only requires mechanical force raises the question that lots of other systems over decades have enabled cleavage of polymer backbones using mechanical force. They did not provide an explanation for why delaying this cleavage would be an advantage in real application.

Nevertheless, I am fine with this manuscript being accepted for publication. It is certainly interesting chemistry.

REVIEWERS' COMMENTS:

Reviewer #1 (Remarks to the Author):

In the revision of their original manuscript, the authors performed a serious amount of additional experiments regarding the initially criticized lack of bulk mechanochemical degradation proof of concept. They used a commercial extruder and blended their mechanochemically responsive BCOE-containing polymers with polycaprolactone, which is fair due to the large amounts that extruders use up. The authors added control experiments showing that the THP-protected polymer does not respond with degradation to the extrusion process and thermal controls show that while the lactonization kinetics, and thus chain scission, are accelerated by heat in the extruder, the initial mechanochemical ring-opening process remained unaffected. Cleverly, the authors use a dye-labelled polymer so the BCOE polymers can be selectively followed by GPC without PCL interference. With these results, the authors clearly demonstrate the feasibility of the degradation process in the bulk and hence give a very promising outlook on the implementation of such a technology in commodity polymers. Though degradation in bulk did not reach the extensive character as compared to the sonication experiments (yet), I agree with the authors here that this is merely a technical issue and I believe such time-consuming iterative optimizations have no place within fundamental cutting-edge research. Overall, I am very much convinced by the excellent work of the authors that addressed all of the issues raised by me earlier and strongly suggest accepting this manuscript for publication in Nature Communications.

Robert Göstl

Response: None needed.

Reviewer #2 (Remarks to the Author):

the authors have properly addressed the comments and this manuscript can be published.

Response: None needed.

Reviewer #3 (Remarks to the Author):

I appreciate the authors' efforts in revising their manuscript. I am not sure how meaningful the new extrusion experiment is: (1) if the polymer degrades upon typical processing, would it be useful? (2) in what application would adding 2.5 wt% of something that degrades be useful? Nevertheless, it is better than nothing.

Response: 1) The extrusion presented here is used to demonstrate the feasibility of BCOE mechanophore in bulk materials. We can use some polymer backbone with relatively low T_g (e.g., $< 25\text{ }^\circ\text{C}$). 2) Because extrusion requires a large amount of sample, we sought this blending strategy, which allows for a reduced sample amount, to demonstrate its utility.

I also found some of the responses to be less than ideal. For example:

1) In response to my comment about the practicality of the approach and casting the work in the context of sustainability, the authors reference a response to a different reviewer focused on bulk material degradation. That is not really what I was asking about, and if anything the extrusion experiment conducted drives home questions about the relevance of this approach.

Response: There are numerous manufacturing methods that require much less (or even no) mechanical processing than is employed here. If a given application required the specific processing conditions employed here, changes to either the polymer matrix or the mechanophore itself could be employed to shift the threshold for activation to higher stresses. We now explicitly point out in the final paragraph that (a) these are proof-of-concept experiments that would need to be optimized for a given application (including those other than sustainability), and (b) specific considerations and possible approaches to that optimization.

2) The response to my next comment about how this work could translate in the future cites other papers but does not answer the question directly. If anything, citing other works detracts from the current work, showing that previous papers have addressed how this might be useful.

Response: We apologize, perhaps we misunderstood the concern and hope that the minor revisions we have made to the introduction and discussion also help to address this.

3) In response to my comment about novelty and the advantages of this mechanism, the authors restate the obvious. Notably, however, their statement that the degradation only requires mechanical force raises the question that lots of other systems over decades have enabled cleavage of polymer backbones using mechanical force. They did not provide an explanation for why delaying this cleavage would be an advantage in real application.

Response: The key advantage here is that whereas conventional mechanical degradation is limited to a single break per polymer stretching event, or between entanglements and/or cross-links in a bulk material, the cascade/delayed lactonization allows for multiple chain scission events within a single stretching event. Thus, there is approximately an order of magnitude greater mechanical degradation than has been observed in any system of which we are aware, other than in the self-immolative polymers reported by Boydston and Moore

and cited in the introduction. We have revised the end of the introduction section to give a little more emphasis to this central aspect of the work.

Nevertheless, I am fine with this manuscript being accepted for publication. It is certainly interesting chemistry.